# Outcomes of Radiofrequency Ablation for Solitary T1a Renal Cell Carcinoma: A 20-Year Tertiary Cancer Center Experience

**DOI:** 10.3390/cancers15030909

**Published:** 2023-01-31

**Authors:** Mohamed E. Abdelsalam, Ahmed Awad, Ali Baiomy, David Irwin, Jose A. Karam, Surena F. Matin, Rahul A. Sheth, Peiman Habibollahi, Bruno C. Odisio, Thomas Lu, Kamran Ahrar

**Affiliations:** 1Department of Interventional Radiology, The University of Texas MD Anderson Cancer Center, Houston, TX 77030, USA; 2Department of Radiology, The University of Texas Southwestern, Dallas, TX 75390, USA; 3Department of Urology, The University of Texas MD Anderson Cancer Center, Houston, TX 77030, USA

**Keywords:** ablation, outcomes, survival rates

## Abstract

**Simple Summary:**

The aim is to evaluate the long-term efficacy and survival of radiofrequency ablation for small renal masses. We reviewed our database over 20 years and concluded that radiofrequency ablation is an effective treatment option of small renal masses. Long-term follow-up revealed a high efficacious treatment modality with low recurrence and complication rates.

**Abstract:**

Background: The aim is to determine the long-term oncologic and survival outcomes of the radiofrequency ablation (RFA) of solitary de novo T1a renal cell carcinoma (RCC). Materials and methods: We retrospectively reviewed our renal ablation registry and included only patients with new solitary, biopsy-proven T1a RCC (<4 cm) who underwent RFA from January 2001 through December 2020. We collected patient and tumor characteristics. Survival rates were estimated using the Kaplan–Meier method. Results: Of the 243 patients who met our inclusion criteria (160 male and 83 female, median age 68 years), 128 (52.6%) had another primary malignancy other than renal malignancy. Two-hundred forty-three RFA procedures were performed for 243 renal tumors of a median tumor size of 2.5 cm. The median follow-up period was 3.7 years. Most tumors (68.6%) were clear cell RCC. Ten patients (4.1%) experienced Clavien–Dindo Grade III complications. Seven patients(3.1%) developed recurrence at the ablation zone, and 11 (4.5%) developed recurrence elsewhere in the kidney. The 15-year local-recurrence- and disease-free survival were 96.5% and 88.6%, respectively. The 15-year metastasis-free survival and cancer-specific survival were 100%. Conclusions: RFA is a highly effective modality for the management of T1a RCC, with low complication and recurrence rates. Long-term data revealed favorable oncologic and survival outcomes.

## 1. Introduction

The advantage of preserving renal function has aroused interest in minimally invasive treatment for small renal masses [1,2,3]. The American Urological Association (AUA) recommends prioritizing partial nephrectomy (PN) for the treatment of small renal masses, rendering conventional radical nephrectomy to specific situations [1]. However, some patients are not candidates for surgery or are not willing to have surgery due to concerns with perioperative complications. Percutaneous image-guided thermal ablation (TA) has emerged as an attractive treatment option for those patients [1,4]. Preservation of renal function is paramount in such patients, as well as those with multiple renal tumors and those with syndromes (e.g., Von Hippel–Lindau) predisposing to renal cell carcinoma (RCC). TA provides local oncologic control in a nephron-sparing fashion. In addition, TA results in fewer complications, shorter recovery, and the possibility of outpatient care compared to surgery [2].

The continuously growing evidence of oncologic efficacy and survival outcomes of the thermal ablation of renal tumors has been recognized by different societies [1,4,5,6]. The recommendations of the American Society of Clinical Oncology (ASCO) clinical practice guidelines include PN as the standard treatment for all patients with small renal masses, with consideration of TA as a treatment option if the lesion is amenable to complete ablation [5]. Furthermore, the National Comprehensive Cancer Network (NCCN) guidelines incorporated ablation as a treatment option for T1a RCC [6]. Since 2017, the American Urological Association (AUA) has recognized thermal ablation as an alternative treatment for lesions <3 cm in selected patients [7].

Radiofrequency ablation (RFA) was introduced in the 1990s as a treatment option for patients with RCC who are not candidates for extirpative surgery. Ever since, the body of literature on its effectiveness and short-term and mid-term outcomes has grown [2,8,9,10,11,12,13,14,15,16,17]. However, fewer data are available on long-term outcomes (more than 5 years) [18,19,20].

To our knowledge, the data on long-term oncologic outcomes and survival rates of percutaneous RFA for T1a RCC is sparse. The purpose of this study was to determine the long-term oncologic effectiveness and survival rates of percutaneous image-guided RFA of biopsy-proven, de novo solitary T1a RCC.

## 2. Materials and Methods

We retrospectively reviewed all the patients in our institutional registry of renal ablations over a 20-year period (January 2001 to December 2020). We obtained Institutional Review Board approval and a waiver of informed consent for this study. All patients with solitary de novo cT1aRCC (≤4 cm) who underwent percutaneous image-guided radiofrequency ablation were included. All RCC diagnoses were based on tissue histology obtained by a biopsy. We excluded patients who underwent cryoablation, patients with syndromes predisposing to RCC (e.g., Von Hippel–Lindau syndrome), patients with a prior history of RCC, patients with metastatic disease from RCC, patients with multiple renal masses, and patients with benign lesions or lesions not biopsy-proven to be RCC.

All patients were referred after multidisciplinary counseling by the treating urologist. An interventional radiologist then evaluated the patients in an outpatient clinic after reviewing the clinical history, imaging studies, and laboratory tests.

### 2.1. Ablation Technique

Our ablation technique was previously described in details [21,22]. In summary, we perform RFA procedures with the patient under general anesthesia using computed tomography (CT) imaging guidance (SOMATOM Definition AS, Siemens Medical Systems, Erlangen, Germany). After ablation planning, the probes are positioned, and we perform the biopsy (if not previously performed) before starting the ablation. Adjunctive techniques (i.e., hydrodissection and pyeloperfusion) are used when needed. When hydrodissection is required, we use dextrose 5% mixed with a nonionic contrast at a ratio of 60:1. We perform RFA for small renal tumors (3.5 cm or smaller) [17] using the Cool-tip RF system (Covidien, Mansfield, MA, USA). The generator is modulated gradually to increase the power up to 150–180 W in order to achieve an ablative temperature of at least 60 °C. Two ablative cycles (6 min each) are applied. Immediately following ablation, we perform multi-phase contrast-enhanced computed tomography (CT) to assess the zone of ablation and identify any immediate complications. Additional cycles of ablation are performed at the discretion of the interventional radiologist if the ablation appears to be incomplete upon immediate post-procedure contrast-enhanced CT. Follow-up cross-sectional imaging is performed at regular intervals for up to 2 years and then every year.

### 2.2. Data Collection

We reviewed the electronic medical record for each patient and recorded the following data: demographics (age and sex), size and laterality of the renal tumor, tumor histology (subtype and Fuhrman grade), history of other non-renal malignancy, technical success of the ablation procedure, thermal ablation technology, imaging guidance modality, adjunctive techniques, complications (graded using the Clavien–Dindo system), residual disease or recurrence at the zone of ablation, tumor recurrence in the kidney (away from the ablation zone), distant metastatic disease from RCC after ablation, whether the patient was alive or dead, as well as the date and cause of death.

### 2.3. Definitions of Outcomes

The procedural and oncologic outcomes are defined according to standardized terminology and reporting criteria published by international image-guided tumor ablation experts [23]. Herein, we define technical success as the successful placement of all the ablation probes and achieving complete ablation. Residual tumor is defined as nodular contrast enhancement identified within the zone of ablation on the first follow-up cross-sectional imaging. Tumor recurrence is defined as a new nodular contrast enhancement within the ablation zone or its margins that was not identified on the prior imaging or viable tumor cells in the ablation zone revealed by tissue sampling. Residual or recurrent tumors were assessed by the urologist and interventional radiologist for all possible treatment options: surgery, active surveillance, or repeat ablation.

Survival outcomes are defined according to the AUA guidelines for the management of small renal masses [24]. Overall survival (OS) represents the proportion of patients alive at the time of data collection. Local-recurrence-free survival (RFS) represents the proportion of patients with no residual or recurrent tumor within the ablation zone. Metastasis-free survival (MFS) describes the proportion of patients who did not develop metastases from RCC in any distant organ. Disease-free survival (DFS) is the proportion of patients with no evidence of RCC disease either in the ablation zone, kidneys (other than the ablation zone), or systemically at the last follow-up cross-sectional imaging. Cancer-specific survival (CSS) describes the proportion of patients who did not die of RCC.

### 2.4. Data Analysis and Statistics

All demographic and tumor characteristics, ablation procedures, complications, and pathologic outcomes were reported using descriptive statistics. The Kaplan–Meier product-limit estimator was used to estimate the OS, RFS, DFS, MFS, and CSS distributions. The OS was calculated from the date of ablation procedure to the date of death. The RFS was calculated from the date of ablation procedure to the date of the diagnosis of recurrence in the zone of ablation. The MFS was calculated from the date of the ablation procedure to the date of the development of metastasis. The DFS was calculated from the date of the ablation procedure to the diagnosis of disease recurrence either in the zone of ablation or elsewhere in the kidney and/or the development of metastasis. The CSS was calculated from the date of the ablation procedure to the date of death caused by RCC disease.

## 3. Results

Two-hundred forty-three patients (one-hundred sixty male and eighty-three female) met our inclusion criteria for this retrospective study. The median age was 68 years (range: 37–87 years). One-hundred twenty-eight patients (52.6%) had another non-renal primary malignancy. The demographics and tumors characteristics are reported in Table 1.

### 3.1. Procedural Outcomes

Two-hundred forty-three ablation procedures for two-hundred forty-three renal tumors were performed in the two-hundred forty-three patients. One-hundred forty renal tumors (57.6%) involved the right kidney, while the rest involved the left kidney. The median size of the renal tumors was 2.5 cm (range: 0.9–3.9 cm). All procedures were performed under CT guidance using RFA. Technical success was 100% as the RF probes’ placement was successful and thermal ablation was achieved in all patients.

Of the 243 ablations, 10 (4.1%) patients developed Grade III or higher Clavien–Dindo complications. Six patients developed bleeding, of whom two required angiography and embolization, two underwent angiography, but no embolization was performed, and two experienced ureteral obstruction requiring percutaneous nephroureteral catheter placement. Two patients developed pneumothorax, of whom one required chest tube placement and the other was treated by percutaneous needle air aspiration. One patient developed a ureteral stricture that was treated with a ureteral stent, and one patient developed infection in the treated kidney that required nephrectomy. The complications and their management are summarized in Table 2.

### 3.2. Pathologic Outcomes

The tissue biopsy of the 243 tumors showed different histological subtypes. The most-common subtype was clear cell RCC (69%), followed by papillary RCC (18.5%), then chromophobe RCC (4.5%). Fuhrman grading was obtained in 204 lesions; most (76%) were Grade II. The pathology subtypes and Fuhrman grades are listed in Table 1.

### 3.3. Oncologic Outcomes

All patients had clinical and radiological follow-up by the Urology and Interventional Radiology teams. Nineteen patients did not have any follow-up imaging after the ablation procedure and were excluded from the analysis. The median follow-up was 3.7 years (range: 0.8–15.4 years). Figure 1 shows a case with a 10-year follow-up imaging.

#### 3.3.1. Residual Disease or Local Recurrence

None of the patients had residual disease on the first follow-up imaging. Local recurrence in the ablation zone was identified in seven patients (3.1%). Twenty-four patients did not show satisfactory involution of the ablated zone upon follow-up cross-sectional imaging; as a result, CT-guided biopsy of the zone of ablation was performed.

Six of the local recurrences were diagnosed by biopsy with no radiological evidence of new enhancement. Only one patient showed radiological criteria of recurrence on follow-up cross-sectional imaging 6 months after the ablation. All local recurrences were diagnosed after at least 6 months of follow-up with a mean time to detection of 8.5 months (range: 6–26.2 months). The tumor histology for the patients with local recurrences was clear cell RCC (n = 5), mucinous and tubular spindle RCC (n = 1), and papillary RCC (n = 1). After multidisciplinary discussion, three of these patients were managed by repeat thermal ablation, three by partial nephrectomy, and one by active surveillance.

#### 3.3.2. Disease Recurrence and Distant Metastases

Eleven patients (4.9%) developed tumor recurrence in the kidneys away from the ablation zone. The median time to detection was 59.5 months. Two of these patients were managed by thermal ablation, and nine patients underwent active surveillance. All patients were free of metastatic disease from RCC at the time of data collection; therefore, the MFS was 100%.

### 3.4. Survival Outcomes

The median OS for the whole patient population in this study was 8.8 years. The 5-, 10-, and 15-year OS rates for all patients in this cohort were 74.7%, 40.7%, and 15.1 %, respectively. Table 3 summarizes the survival rates.

The median LRFS and DFS were 8.39 and 8.24 years, respectively. The LRFS rate was 96.5% at 5, 10, and 15 years. The DFS rate was 92.3% at 5 and 88.6% at 10 and 15 years. The MFS, as well as CSS were 100%, as none of the patients developed distant metastatic disease or died of RCC. Figure 2 shows the Kaplan–Meier curves of the OS, LRFS, and DFS.

## 4. Discussion

The AUA guidelines recommend prioritizing partial nephrectomy (PN) for the management of the cT1a renal mass and considering thermal ablation as an alternative approach for the management of cT1a renal masses 3 cm or less in size [1]. In clinical practice, minimally invasive image-guided ablation of small renal tumors has gained wide acceptance [8,9,10,11,12,13,14,15,16,18,19,25]. In our previously reported experience with renal ablation, the authors reported all patients with renal tumors who underwent laparoscopic or percutaneous ablation, including those patients with familial genetic syndromes, metastatic disease, and those with multiple tumors or non-histology-proven RCCs [10]. The current study reports an experience with a more homogenous patient population who underwent only percutaneous image-guided radiofrequency ablation for solitary de novo biopsy-proven T1a RCCs. To the best of our knowledge, the current study reports the most extensive experience with the largest patient population of percutaneous RFA for biopsy-proven T1a RCCs. The current study details our 20 years of experience in a large cancer center with a total of 243 patients and a median follow-up of 3.7 yr (range: 0.8–15.4).

This study demonstrates the 5-year, 10-year, and 15-year OS rates for all patients of 74.7%, 40.7%, and 15.1% respectively. In a prior study [26], we identified a statically significant difference in the OS between patients who had RCC only versus those who had RCC and another non-renal primary malignancy. This explains the OS rates in the current study, given that 53% of the patients had another non-renal malignancy. However, the median OS for the whole group in this study was 8.8 years.

Psutka et al. [18] shared their experience of RFA for T1a and T1b RCC lesions. This study included only patients with solitary de novo histology-proven RCC. Patients with prior RCC, multiple lesions, lesions bigger than 7 cm, and those with familial syndromes were excluded. In the subgroup of patients with T1a RCC (n = 143), the authors reported a 5-year local RFS of 96.1% and a 10-year local RFS of 93.2%. This is in concordance with the local RFS in our current study of 96.5% at 5, 10, and 15 years. Furthermore, those authors reported a 5-year DFS of 91.5% [18]. Our 5-, 10-, and 15-year DFS was 92.3%, 88.6%, and 88.6%, respectively. In addition, our 100% MFS and CSS rates are similar to their reported rates [18].

Wah et al. [19] published the outcomes of the RFA of 200 renal tumors (183 were RCC) with a median follow-up of 3.8 years. The authors reported 5-year OS, CSS, LRFS, and MFS rates of 75.8%, 97.9%, 93.5%, and 87.7%, respectively, for the entire population [19]. Ma et al. [25] reported the long-term oncological outcome of RFA for small renal masses. The study included 52 patients with a median follow-up of 5 years. Of the 58 ablated renal tumors, 41 lesions were biopsy-proven to be RCC. The authors reported a 5- and 10-year RFS of 94.2%. The MFS and CSS were 100% as none of the patients developed metastasis or died from RCC during the follow-up period [25]. In a more recent study, Johnson et al. [20] evaluated the long-term outcome of RFA in 102 patients with a median follow-up of 6.6 years. Sixty-two lesions were biopsy-proven to be RCC. The 6-year DFS, MFS, and CSS were 89%, 96%, and 96%, respectively. In a subgroup analysis of patients with at least 10 years of follow-up imaging, the respective OS, DFS, and CSS were 49%, 82%, and 94%. The reported rates are in agreement with our reported survival rates.

Our survival rates are within the range of the 5-year [27,28] and 10-year survival rates [29,30] reported for PN. Andrews et al. [27] shared their experience and reported 5-year survival rates for T1a RCC treated with PN for 1055 patients (835 were biopsy-proven RCC) with a median follow-up of 9.4 years. The authors reported an LRFS, MFS, and CSS of 97.4%, 98%, and 99.3%, respectively [27]. Antonelli et al. reported their experience with PN for T1a RCC in 992 patients. The 5- and 10-year CSS were 96.1% and 94.9%, respectively [29]. Xing et al. [31] reported the OS and CSS for 691 patients who underwent PN for T1aRCC. The 5- and 9-year CSS were 97.4% and 96.4%, respectively [31].

We acknowledge the strengths and the limitations of our study. Our study population is larger than those of the comparable studies in the existing literature. Another strength is the long follow-up time (85 patients had a follow-up of more than 5 years). Furthermore, only patients with biopsy-proven RCC were included in this study. In addition, all the procedures were performed percutaneously by three interventional radiologists, currently with 21-, 11-, and 7-years of experience. This promoted consistency in the technique and decreased technical variations. A limitation is that the current study reflects a single-institution cancer center experience; a multi-institution study with a bigger sample more representative of the overall patient population would have increased the power of the study. Another limitation of the study is its retrospective design with all the inherent limitations, e.g., patient selection. Furthermore, this was a single-arm study with no control group to compare the outcomes of ablation with those of a non-ablation group. Future prospective randomized control studies are needed to confirm the current results and allow head-to-head comparisons between different treatment options for T1aRCC.

## 5. Conclusions

In conclusion, RFA is a highly efficacious modality and provides an effective durable treatment option for patients with T1a RCC. The long-term data reveal low complication and recurrence rates with favorable long-term oncologic control and survival outcomes.

## Figures and Tables

**Figure 1 cancers-15-00909-f001:**
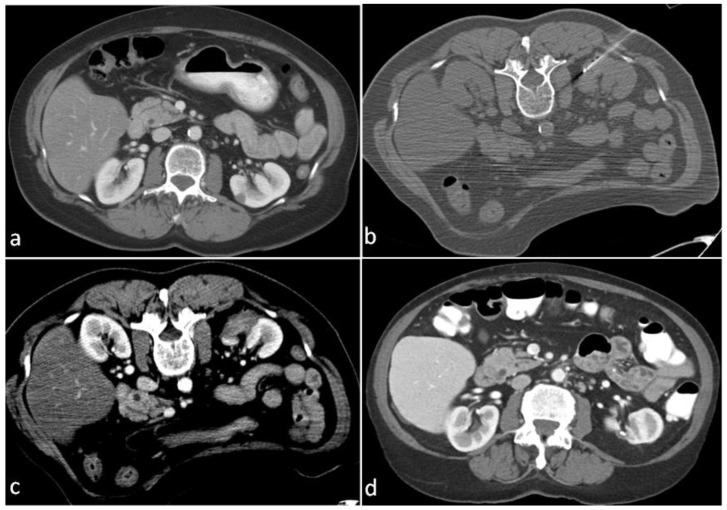
A 75-year-old female with a history of breast cancer. (**a**) Contrast-enhanced CT during surveillance revealed a 1.2 cm left renal mass. Biopsy revealed renal cell carcinoma, papillary Type 1, Fuhrman’s nuclear Grade 2. (**b**) The lesion was treated with CT-guided radiofrequency ablation. (**c**) Contrast-enhanced CT immediately after the ablation revealed a lack of enhancement delineating the margins of the ablation zone. (**d**) Contrast-enhanced CT 10 years following the ablation shows the resolution of the ablation zone and the development of dystrophic calcification.

**Figure 2 cancers-15-00909-f002:**
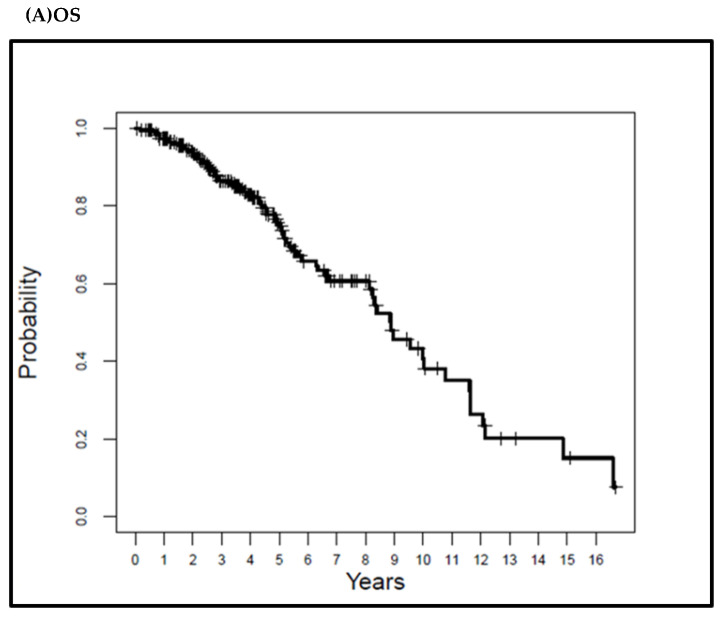
Kaplan–Meier overall survival (**A**), local-recurrence-free survival (**B**), and disease-free survival (**C**) curves for the study patients.

**Table 1 cancers-15-00909-t001:** Patients and tumor characteristics (N = 243).

Characteristic	n	%
Age (years)		
Mean (SD)	67.0 (11.4)	
Median (range)	67.7 (37.6–87.4)	
Gender		
Female	83	34.2
Male	160	65.8
Another primary malignancy		
Yes	128	52.7
No	115	47.3
Size of lesion (cm)		
Mean (SD)	2.4 (0.6)	
Median (range)	2.5 (0.9–3.9)	
Affected kidney		
Left	103	42.4
Right	140	57.6
Ablation modality		
RFA	255	100
Guidance modality		
CT	255	100
Pathology		
RCC, chromophobe	11	4.5
RCC, clear cell	175	72.0
RCC, papillary	45	18.5
RCC, not otherwise specified	5	2.1
RCC, mucinous and tubular and spindle	3	1.2
RCC, clear cell papillary	2	0.8
RCC, papillary versus clear cell papillary	1	0.4
RCC, papillary versus mucinous and tubular	1	0.4
Grade	N = 204	
1	36	17.6
2	155	76
3	13	6.4

**Table 2 cancers-15-00909-t002:** Clavien–Dindo Grade III complications and their management.

Complication	Management
Subcapsular hematoma	Angiography and embolization
Hematuria, obstructing clot	Percutaneous nephroureteral catheter
Subcapsular hematoma	Angiography (no embolization)
Pneumothorax	Chest tube
Perirenal hematoma	Angiography (no embolization)
Bleeding, obstructing clot	Percutaneous nephroureteral catheter
Pneumothorax	Aspiration
Subcapsular hematoma	Angiography and embolization
Renal infection	Nephrectomy
Ureteral stricture	Ureteral stent

**Table 3 cancers-15-00909-t003:** Survival rates of the study patients.

	Median (Years)	5-Year Survival (%) (95% CI)	10-Year Survival (%) (95% CI)	15-Year Survival (%) (95% CI)
OS	8.88	74.7 (67.9–82.3)	40.7 (30–55.3)	15.1 (6.5–35)
LRFS	8.39	96.5 (94–99.1)	96.5 (94–99.1)	96.5 (94–99.1)
DFS	8.24	92.3 (88.6–96.2)	88.6 (82.7–95)	88.6 (82.7–95)

OS, overall survival; LRFS, local-recurrence-free survival; DFS, disease-free survival.

## Data Availability

The datasets generated during and/or analyzed during the current study are available from the corresponding author upon reasonable request.

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
