# Peer review of "Outcomes of Radiofrequency Ablation for Solitary T1a Renal Cell Carcinoma: A 20-Year Tertiary Cancer Center Experience"

_cancers, 2023, doi:10.3390/cancers15030909_

Round 1
Reviewer 1 Report
Overall, this is an interesting study, well-written and easy to follow. I have some minor comments:
1) It would be nice to see some pre- post- and follow-up CT images of representative/interesting cases.
2) 20 years is a very long period of time and changes in treatment protocol would be expected. Furthermore, a trend over time (e.g., complications were more common in first decade compared to second? survival got better for treatments performed in the second decade compared to the first?) might be observed and interesting for readers. It might also be possible to assess some sort of learning curve.
3) More details should be provided regarding the average number of operators and their experience in performing the procedure during the 20 years on average at least.
Author Response
Overall, this is an interesting study, well-written and easy to follow. I have some minor comments:
1) It would be nice to see some pre- post- and follow-up CT images of representative/interesting cases.
Thank you so much. Changes have been made to the article.
2) 20 years is a very long period of time and changes in treatment protocol would be expected. Furthermore, a trend over time (e.g., complications were more common in first decade compared to second? survival got better for treatments performed in the second decade compared to the first?) might be observed and interesting for readers. It might also be possible to assess some sort of learning curve.
Thank you so much for your comment. All the procedures were performed by three interventional radiologists. We are using the same technique established by the most senior interventional radiologist and published back in 2005. The complication and the recurrence rates are very low that we were not able to compare the first and 2nd decade rates. Given the comorbidities of this patient population as well as the 53% of patients have another primary cancer other than RCC, the OS survival is affected by these 2 factors as we published in the past. However, the RCC cancer specific survival was not different between the 1st and 2nd decade as none of the patients died from RCC.
3) More details should be provided regarding the average number of operators and their experience in performing the procedure during the 20 years on average at least.
Thank you so much. All the procedures were performed by three interventional radiologists, currently with 21, 10- and 7-years’ experience. Changes have been made to the article.
Reviewer 2 Report
I read with great interest the authors manuscript. They analyzed a large cohort of renal tumors treated with radifrequency ablation that were followed in a median of 3.7 years.
I have just one major concern with 3.7 years of follow up you can not state 10 and 15 years; LRFS, local recurrence–free survival; DFS, disease-free survival, of course, estimates but they do not reflect the reality. The reality is that 6 out of 10 are not reaching 10 years of survival in a poupulation with a mean age of 67.
Major revision after careful modification of the result (maybe only for the 85 patients how to have more than 5 years of follow-up) and acknowledgment of limitations.
Author Response
I read with great interest the authors manuscript. They analyzed a large cohort of renal tumors treated with radifrequency ablation that were followed in a median of 3.7 years.
I have just one major concern with 3.7 years of follow up you can not state 10 and 15 years; LRFS, local recurrence–free survival; DFS, disease-free survival, of course, estimates but they do not reflect the reality. The reality is that 6 out of 10 are not reaching 10 years of survival in a poupulation with a mean age of 67.
Major revision after careful modification of the result (maybe only for the 85 patients how to have more than 5 years of follow-up) and acknowledgment of limitations.
Response: Thank you so much for your comment. We reported the data using the same way it’s been reported in the urology literature. This consistency in reporting helps with the comparison between the data in interventional radiology and the urology literature, which in turn is in the favor of the patient care. The following 2 articles are examples for reporting from the urology literature.
Campbell SC, Novick AC, Belldegrun A, Blute ML, Chow GK, Derweesh IH, et al. Guideline for management of the clinical T1 renal mass. The Journal of urology. 2009;182(4):1271-9.
Psutka SP, Feldman AS, McDougal WS, McGovern FJ, Mueller P, Gervais DA. Long-term oncologic outcomes after radiofrequency ablation for T1 renal cell carcinoma. European urology. 2013;63(3):486-92.
Round 2
Reviewer 1 Report
The Authors satisfactorily addressed my comments and I have no additional remarks
Reviewer 2 Report
Good modifications